# Impact of prolonged infection on SARS-CoV-2 evolution

Fang Yan,[1,2] Qiushi Jin,[2,3] Yuanguo Li,[2] Xuefeng Wang,[2] Chunling Dong,[4] Xianzhu Xia,[2] Yuwei Gao,[2] Jie Zhang,[4] Zhijun Hou[1]

**ABSTRACT** Immunocompromised patients with prolonged severe acute respiratory syndrome coronavirus 2 (SARS-CoV-2) infections may serve as reservoirs for viral evolution, with suboptimal immune responses facilitating the accumulation of adaptive mutations. This study aimed to characterize the drivers of SARS-CoV-2 adaptive evolution in such hosts through genomic surveillance. We retrospectively analyzed 24 patients with long-term positive nasopharyngeal reverse transcription-polymerase chain reaction results (symptom onset duration: 7–14 days on average, 1 HIV-positive patient with >20 days of infection). Most infections (April–May 2022) were caused by Omicron variants (predominantly BA.2). Phylogenetic analysis revealed accelerated viral evolution in patients with diverse underlying diseases (e.g., HIV and esophageal cancer). A total of 78 intrahost single-nucleotide variants were identified, with ORF1ab (53.8%) and the Spike protein coding region (20.5%) being hotspots. Notably, the HIV-positive patient's virus developed unique mutations: NSP3-T779I, NSP15-A94T, and Spike double mutations N440K and I794T. Functional assays showed that the N440K/I794T double mutation significantly enhanced infectivity in Hela-hACE2 cells ($P < 0.05$) but reduced immune evasion (50% neutralizing titer increased ~2-fold vs BA.2, $P < 0.001$). The I794T mutation was later detected in the JN.1.16 strain, suggesting potential evolutionary persistence. Prolonged SARS-CoV-2 replication in immunocompromised hosts, particularly HIV-positive individuals, drives adaptive mutations with altered infectivity and immune evasion. These findings emphasize the need for monitoring such hosts to prevent the spread of potentially transmissible variants.

**IMPORTANCE** This study reveals the characteristics of adaptive mutations and their biological functions of SARS-CoV-2 during long-term infection in immunocompromised patients, emphasizes the importance of close monitoring of the long-term replication of SARS-CoV-2, aiming to timely detect and prevent the spread of newly emerging neutralization-resistant variants in susceptible populations, and provides a key scientific basis for formulating effective public health prevention and control strategies.

**KEYWORDS** accelerated viral evolution in immunocompromised hosts, hotspots of intrahost mutations, unique mutations in the HIV-positive patient

The severe acute respiratory syndrome coronavirus 2 (SARS-CoV-2) pandemic has been characterized by the regular emergence of new variants. The origin of these new variants is unclear, with some speculating emergence from zoonotic spillover into other vertebrates and spillback into humans (1–5). An alternative, widely debated source of new variants stems from persistent infections in immunocompromised patients. In such hosts, suboptimal immune responses are supposed to facilitate the gradual accumulation of genomic adaptations (4–7), which probably occurred prior to the major SARS-CoV-2 wave in December 2022–January 2023, when China's population exhibited a distinct immunological profile characterized by lower population immunity due to minimal prior infection exposure. This contrasted sharply with nations that had experienced repeated infection waves. China's unique immunological landscape,

**Peer Reviewer** Sushanta Deb, AIIMS, Bhubaneswar, Odisha, India

Address correspondence to Yuwei Gao, dawei1105@foxmail.com, Jie Zhang, doctorzhangj@163.com, or Zhijun Hou, houzhijundb@163.com.

Fang Yan, Qiushi Jin, and Yuanguo Li contributed equally to this article. Author order was determined by drawing straws.

Yuwei Gao, Jie Zhang, and Zhijun Hou contributed equally to this article.

The authors declare no conflict of interest.

coupled with prolonged circulation of similar viral strains both domestically and globally, provides a rare opportunity to systematically examine viral evolutionary patterns across divergent immune milieus (8). SARS-CoV-2 infections typically resolve clinically within days, with RNA shedding persisting from several days to weeks (9). However, accumulating case reports describe chronic infections lasting weeks to months, challenging this conventional understanding of viral clearance dynamics (10, 11). Chronic SARS-CoV-2 infections are characterized by prolonged detection of replicative virus. To date, all documented cases have occurred exclusively in severely immunocompromised individuals, including those with primary immunodeficiencies, post-transplant immuno-suppressive therapy, AIDS, hematological malignancies, and/or associated treatments.

The persistence of replicative virus in these patients likely results from impaired viral clearance mechanisms, particularly involving adaptive immune dysfunction. This contrasts sharply with immunocompetent individuals, who typically clear the infection efficiently (12).

Longitudinal sequencing analyses of chronic infection cases have revealed a greater number of mutations and distinctive mutational patterns compared to those observed in transmission chains among acutely infected individuals (6, 13, 14). Multiple studies have characterized viral evolutionary dynamics in immunocompromised hosts, demonstrat-ing a strong association between immunodeficiency and accelerated within-host viral mutation accumulation (6, 7, 13–21). A recent study revealed that 13.9% of SARS-CoV-2-infected B-cell lymphoma patients experienced prolonged infections persisting ≥30 days (22). Patients with B-cell lymphoma often exhibit impaired SARS-CoV-2-neutraliz-ing antibody production, which predisposes them to both prolonged infection (23–25) and diminished vaccine responsiveness (26, 27). Notably, one B-cell lymphoma patient maintained persistent SARS-CoV-2 infection for 156 days, during which the virus acquired 16 mutations, including 4 substitutions conferring neutralizing antibody escape (28). In a separate case series, multiple immunocompromised heart transplant recipients independently acquired the immune-evading E484K Spike substitution within a remarkably short 14-day period (18). Current evidence suggests that the predominant selective pressure in these cases likely stems from adaptive advantages conferred by enhanced cell-to-cell transmission within the host (29). Emerging evidence suggests that immunocompromised hosts may serve as reservoirs for accelerated SARS-CoV-2 variant evolution, though the precise dynamics and frequency of this phenomenon remain to be systematically quantified.

In this study, we conducted genomic surveillance of SARS-CoV-2 in 24 immunocom-promised patients with suspected chronic infection to track the emergence of intrahost single-nucleotide variants (s). Through longitudinal sequence analysis, we characterized viral evolutionary dynamics within these hosts and compared mutation accumulation rates with those observed in immunocompetent populations.

## RESULTS

### Adaptive evolutionary characteristics in long-term SARS-CoV-2 infections

In immunocompromised patients with long-term chronic severe acute respiratory syndrome coronavirus 2 (SARS-CoV-2) infection, the virus often accumulates a large number of adaptive mutations due to continuous replication. These mutations may significantly alter its pathogenic characteristics, including infectivity, pathogenicity, and immune escape ability. This study focuses on this important scientific issue and conducts a retrospective analysis of 24 early patients whose nasopharyngeal reverse transcription quantitative polymerase chain reaction (RT-qPCR) tests remained positive for a long time. The 24 patients are aged between 16 and 70 years old. In this study, we define chronic infection as the persistence of positive SARS-CoV-2 RNA in nasopharyngeal samples from patients more than 7 days after the onset of symptoms. These patients have multiple underlying diseases, including HIV, esophageal cancer, heart disease, cerebral infarction, hypertension, and diabetes mellitus (Table 1). The diverse background of underlying

**TABLE 1** Summary of all 27 patients with prolonged SARS-CoV-2 infections[a,b]

| Acquisition time | Sample number | Background condition | CT:N | Lineages | Number of intrahost amino acid variants |
|---|---|---|---|---|---|
| 19 April 2022 | 1 | Short bowel syndrome, coronary heart disease, atrial fibrillation | 37 | / | / |
| 19 April 2022 | 2 | HIV | 90 | BA.2 | 2.00 |
| 19 April 2022 | 3 | Esophageal carcinoma | 73 | / | / |
| 19 April 2022 | 4 | / | 61 | BA.2 | / |
| 19 April 2022 | 5 | / | 31.00 | / | / |
| 19 April 2022 | 6 | / | 20 | BA.2 | / |
| 20 April 2022 | 7 | / | 00 | BA.2 | / |
| 20 April 2022 | 8 | Diabetes | 02 | BA.2 | / |
| 22 April 2022 | 9 | Hypertension | 21 | BA.2 | / |
| 23 April 2022 | 10 | / | 00 | BA.2 | / |
| 23 April 2022 | 11 | / | 67 | BA.2 | / |
| 21 April 2022 | 12 | / | 87 | BA.2 | / |
| 22 April 2022 | 13 | / | 00 | BA.2 | / |
| 22 April 2022 | 14 | / | 64 | B.1.1.529 | / |
| 22 April 2022 | 15 | / | 00 | BA.2 | / |
| 22 April 2022 | 16 | / | 00 | B | / |
| 22 April 2022 | 17 | / | 00 | / | / |
| 4 May 2022 | 18 | / | 14 | / | / |
| 4 May 2022 | 19 | Cerebral infarction | 59 | / | / |
| 4 May 2022 | 20 | / | 34 | B.1.1 | / |
| 4 May 2022 | 21 | / | 01 | / | / |
| 4 May 2022 | 22 | / | 14 | / | / |
| 4 May 2022 | 23 | / | 80 | / | / |
| 4 May 2022 | 24 | / | 00 | / | / |

[a]Number of intrahost amino acid variants is the result of comparison with the BA.2 reference sequence during the same period.
[b]/, not detected.

diseases provides a rich sample for exploring the impact of different immune statuses on viral infection.

We performed RT-qPCR and deep sequencing on 14-day samples from each patient, with all selected samples having a cycle threshold value of <32, among which 12 samples were identified as Omicron strains (Table 1). During the period of April–May 2022 when the study was conducted, the dominant circulating SARS-CoV-2 strains in China were BA.1/BA.2 (Fig. 2A, data sourced from Global Initiative on Sharing All Influenza Data [GISAID]). However, phylogenetic tree analysis of 14 patients in this study cohort revealed that under different immune backgrounds, the virus accumulated various mutations during the prolonged infection period, which accelerated the viral evolutionary rate (Fig. 1). Notably, we also identified two WH-01-Hu-like strains, and this finding indicates that due to differences in patients' immune backgrounds, the virus exhibits a greater number of mutations during infection.

To further explore the virus's activity and replication capacity, we attempted to isolate the virus from these clinical samples. The results showed that only the virus from HIV patients' samples could be successfully isolated and amplified in Vero E6-TMPRSS2 cells. The viral sgRNA copy number detected by RT-qPCR and the virus titer determined by PFU were $3 \times 10^5$ (Fig. 2B and C). Based on this, we hypothesize that the number of viable virus particles in immunosuppressed patients is higher, making them more likely

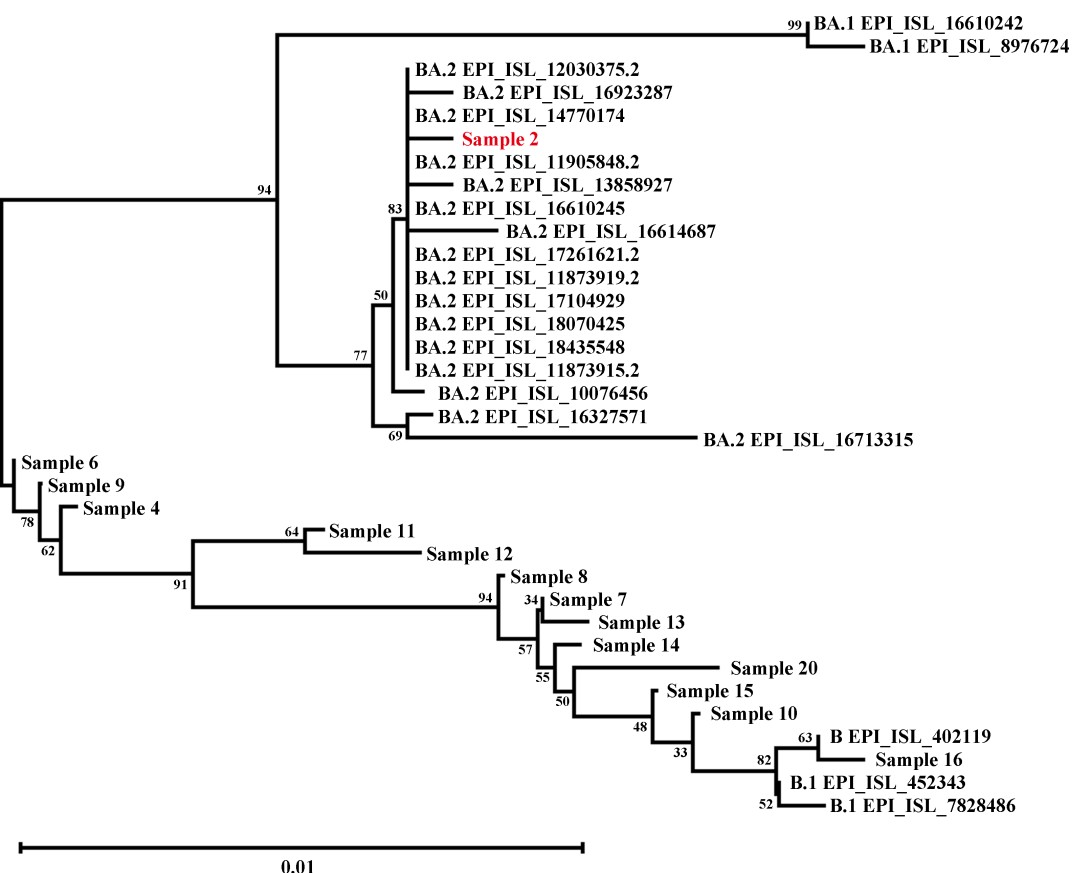

**FIG 1** Viral evolutionary dynamics in prolonged SARS-CoV-2 infections. Maximum-likelihood phylogenetic tree of 14 clinical isolates, with sample 2 highlighted as originating from an HIV-positive patient. Branch lengths represent nucleotide substitutions per site. Scale bar indicates genetic distance.

to persist for a long time. This is consistent with reports in multiple previous studies that SARS-CoV-2 can chronically infect HIV patients for more than 100 days.

## Intrahost genetic diversity during chronic SARS-CoV-2 infection

Through systematic genetic screening of full-genome mutations, a total of 78 mutant/deletion iSNVs were identified. The allele frequencies of these iSNVs range from 5% to 98% (with no variant reaching 100% fixation) and can be further categorized based on within-host dynamic characteristics: 29.5% (23/78) are low-frequency expanding variants (5%–30%); 61.5% (48/78) are moderate-frequency stable variants (30%–90%); and 9.0% (7/78) are high-frequency unfixed variants (90%–98%). This distribution indicates that all iSNVs are in the dynamic evolutionary stage within the host (not fully fixed), consistent with the characteristics of adaptive mutations accumulating during prolonged infection (Table S1). Further genomic localization analysis showed that these iSNVs are widely distributed across nine open reading frames (ORFs) of SARS-CoV-2 (Fig. 3). To eliminate the influence of ORF length on variant count, we calculated the variant density (number of iSNVs per 1,000 nt) for each ORF and verified differences via Poisson regression (Table S1). Results showed that (i) in terms of absolute iSNV count, ORF1ab had the highest number of variants (42 sites, 53.8% of total) due to its longest reference length (21,552 nt); (ii) in terms of variant density, ORF6 (13.16 sites/1,000 nt; RR = 6.75; $P = 0.002$), N (6.35 sites/1,000 nt; RR = 3.26; $P = 0.002$), and Spike (4.19 sites/1,000 nt; RR = 2.15; $P = 0.007$) had significantly higher densities than ORF1ab (1.95 sites/1,000 nt), while ORF3a, ORF7a, ORF8, and E showed no significant differences (all $P > 0.05$). This suggests that short ORFs like ORF6, N, and Spike may be under stronger selection pressure during prolonged infection, leading to higher variant accumulation per unit length.

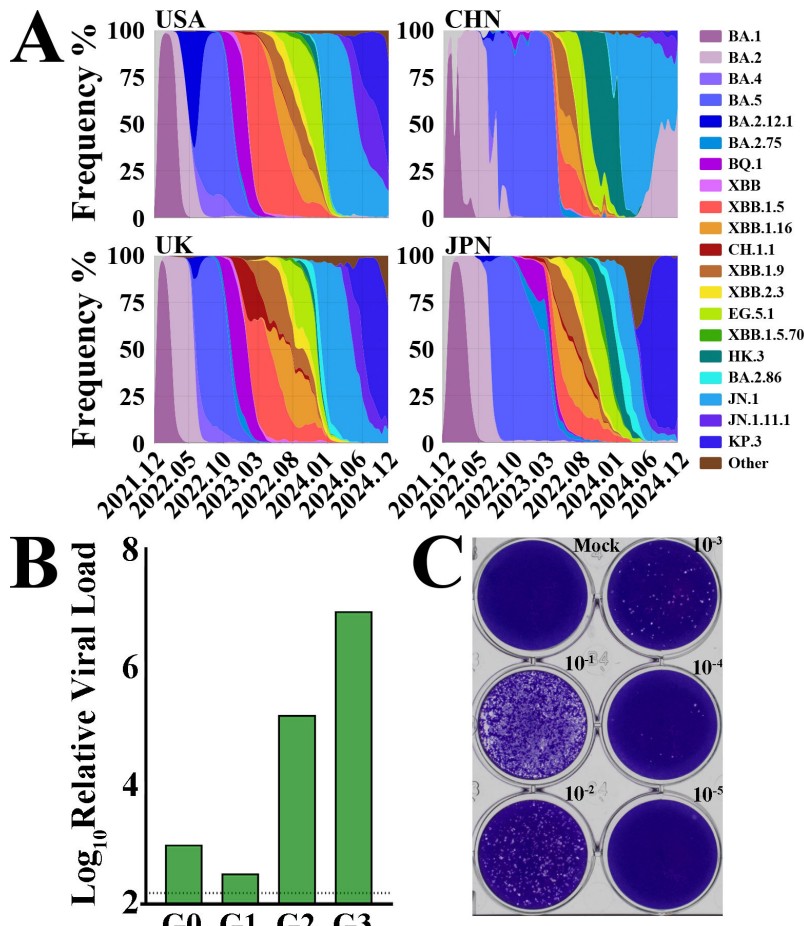

**FIG 2** Viral isolation and characterization from clinical samples. (A) SARS-CoV-2 epidemiological trends (January–December 2022). Prevalence of Omicron sublineages in the United States (USA), China (CHN), the United Kingdom of Great Britain and Northern Ireland (UK), and Japan (JPN) for 3 years from December 2023 (2023.01) to December 2024 (2024.03). (B) Viral load quantification by RT-qPCR. G0, original clinical sample. (C) Plaque assay of HIV-patient isolates in Vero E6-TMPRSS2 cells. The virus was serially diluted (10-fold increments).

Among the above variants, mutations at specific sites show high occurrence frequency and abundance, suggesting that they may have potential selective advantages in the process of virus adaptation to the host or replication. Among them, the G28423T mutation site in the N protein coding region has the highest occurrence

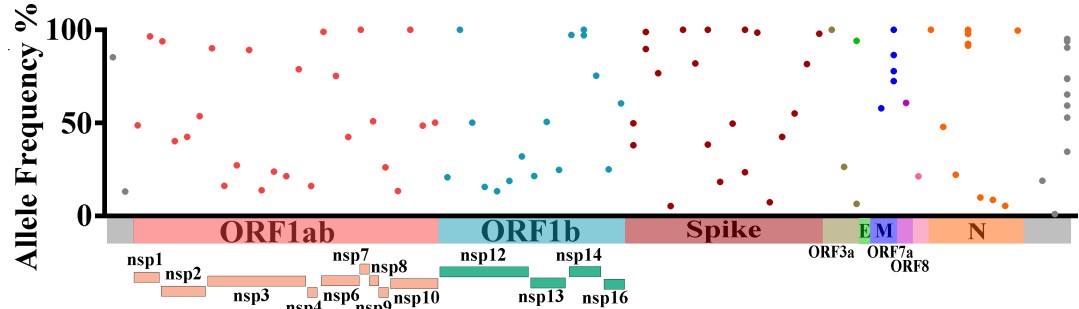

**FIG 3** Intrahost genetic assessment of the genome of long-term SARS-CoV-2 patients. Genomic distribution and frequency of intrahost single-nucleotide variants. ORF1ab (pink), ORF1b (cyan), Spike (red), ORF3a (yellow), E (green), M (blue), ORF7a (purple), ORF8 (light pink), and N (orange).

frequency, reaching 10 times, and its allele frequency is maintained above 91% (>91%) but not reaching 100%, indicating that this mutation is nearly fixed in the tested samples and may have a strong adaptive advantage. It is followed by the C29870A mutation in the 3′ untranslated region, which occurs nine times with a wide frequency range (>30% and <95%), reflecting that this site may have differential evolutionary dynamics in different samples. The T26843C mutation in the M protein coding region occurs four times, with a frequency ranging from 70% to 100%, suggesting that it has become a dominant variant in some samples.

iSNVs in the Spike protein coding region and the non-structural protein 14 (NSP14) coding region each occurred twice. Specifically, the C18692T mutation in the NSP14 coding region had frequencies of 97% and 100% in the two detections, indicating that this mutation is nearly completely fixed in the corresponding samples. Mutations in the S protein coding region exhibit diverse frequency characteristics: the frequencies of the A22301C mutation are 49.76% and 38.03%; those of C22480T are 89.63% and 98.82%; those of A23251G are 38.32% and 100%; and those of A24531C are 23.52% and 100%. Notably, mutation sites such as C18692T (NSP14), A22301C (Spike), C22480T (Spike), A23251G (Spike), A24531C (Spike), T26843C (membrane protein), and G28423T (nucleocapsid protein) have become key marker sites for defining subsequent circulating strains due to their multiple occurrences and high frequencies in some samples.

Through comparative analysis with clinical isolates from the same period, it was found that the number of nucleotide and amino acid variations in the S protein of samples from HIV-coinfected patients was significantly higher than that in ordinary clinical isolates. This result is consistent with reports from multiple previous studies, further confirming that the evolutionary rate of the virus is significantly accelerated in the context of immunocompromised hosts (Fig. S1 and Table 1). In addition, deep sequencing results of sample 2 showed that immunosuppressed patients may not only experience persistent long-term infections but also be accompanied by the phenomenon of "hypermutation" in the viral genome, providing important empirical evidence for understanding the adaptive evolutionary mechanism of the virus under immunodeficiency.

## Genomic characterization of persistent SARS-CoV-2 infection in immunocompromised patients

To explore the evolutionary characteristics of the SARS-CoV-2 BA.2 subtype during a specific period, this study systematically retrieved the viral evolutionary patterns of all BA.2-infected patients in China from 1 February 2022 to 30 June 2022 and focused on comparative analysis of the viral evolutionary pattern of patient no. 2 with that of patients in Jilin Province during the same period (Fig. 4A). Through sequence alignment and variation analysis, we identified six iSNVs in the viral genome of patient no. 2 (Fig. 4B). Further functional annotation showed that four of these variations were non-synonymous substitutions, located at the T779I site of NSP3, the A94T site of NSP15, and the N440K and I794T sites of the Spike protein, with their substitution frequencies in the viral population being 13.9%, 100.0%, 5.37%, and 18.35% in sequence (Fig. 4B). Notably, these variation patterns are not entirely consistent with the evolutionary characteristics presented by SARS-CoV-2 in the early stage of transmission, suggesting that the virus may exhibit a unique evolutionary path under specific host environments or infection conditions. To further clarify the specificity and universality of the above four non-synonymous substitutions, we extensively compared them with viral sequences from other domestic BA.2-infected patients during the same period. The results showed that substitutions at the NSP3-T779I and Spike protein (I794T) sites were not detected in domestic patients during the same period, indicating that they may be unique adaptive variations of patient no. 2; the substitution at NSP15-A94T appeared in some patients during the same period but with a low frequency, accounting for only 18.67%; while the Spike protein (N440K), as a defining site of the BA.2 subtype, underwent a reversion mutation in patient no. 2 (Fig. 4B). This phenomenon suggests that the variation at

this site may be dynamically regulated by host immune pressure or viral replication requirements.

To verify whether the aforementioned variations stably exist in the viral population, we performed plaque purification on the viral strain isolated from patient no. 2 and conducted genome sequencing on the purified monoclonal virus. The results showed that none of the aforementioned mutations were detected in the purified viral clones, which suggests that these mutations generated by the virus may be in an unstable state in patients with intact immune function. They cannot survive stably or proliferate effectively in the *in vitro* culture environment of the Vero E6-TMPRSS2 cell line. It is speculated that such mutations may be closely related to the host's immune selection pressure and are difficult to maintain in cell culture systems lacking corresponding selection pressure. Notably, it is of research value that we found the S protein (I794T) mutation site was detected in the subsequently isolated clinical strain JN.1.16. This finding implies that this mutation may have certain evolutionary potential, being retained and stably inherited in specific transmission chains or evolutionary branches, thus providing important clues for exploring the evolutionary trajectory of the virus.

Statistical analysis of the coding region distribution of amino acid variations in the viral genome of patient no. 2 showed that the coding regions with the largest number of variations were, in order, the Spike protein, nucleocapsid protein, and membrane envelope protein (Fig. 4C). Among them, the Spike protein had as many as 31 mutations, the nucleocapsid protein had seven mutations, and the membrane protein had three mutations. As a key structure for the virus to bind to host cell receptors, the high-frequency mutations in the Spike protein may be related to adaptive evolutionary strategies such as the virus evading host immune recognition and enhancing infection efficiency.

Synthesizing the above research results, we hypothesize that during chronic infection, the evolution of the virus may face a trade-off between multiple selection pressures. This trade-off effect may involve multiple aspects such as viral replication efficiency, immune escape ability, and host adaptability, thereby affecting the ability of viral variants generated during chronic infection to further spread and persistently replicate in the population. This finding provides important experimental basis and theoretical reference for an in-depth understanding of the evolutionary mechanism of SARS-CoV-2 in chronically infected hosts and evaluating the transmission risk of potential variants.

## SARS-CoV-2 Spike mutations confer enhanced infectivity and antibody neutralization escape

To further investigate the impact of specific mutations in the Spike protein of the SARS-CoV-2 BA.2 subtype on viral infectivity and immune escape ability, this study employed a vesicular stomatitis virus (VSV) pseudotyping system for functional verification. We constructed six types of pseudotyped viruses, which express the wild-type D614G mutant Spike protein (WT-Spike-D614G), BA.2 subtype Spike protein (BA.2-Spike), BA.2 subtype N440 mutant Spike protein, BA.2 subtype T794 mutant Spike protein, and BA.2 subtype N440/T794 double-mutant Spike protein BA.2-Spike-N440/T794, respectively. By detecting the neutralizing activity of serum against these pseudotyped viruses, we systematically evaluated the functional effects of the mutation sites. This pseudotyped virus system has been proven to be widely used in studies on viral entry mechanisms and neutralizing antibody evaluation, and its detection results are in good agreement with those of experiments using fully replication-competent viruses. In the detection of viral infectivity, we used reverse transcriptase activity as a quantitative indicator to assess the single-round infection ability of pseudotyped viruses in Hela-hACE2 cells. The results showed that compared with WT-Spike-D614G, the single-round infectivity of BA.2-Spike in Hela-hACE2 cells was significantly reduced, while the single-round infectivity of the BA.2-Spike-N440/T794 double mutant was significantly enhanced, with a statistically significant difference compared with BA.2-Spike ($P < 0.05$), suggesting that the synergistic effect of the double mutation can significantly improve the cell invasion ability of the

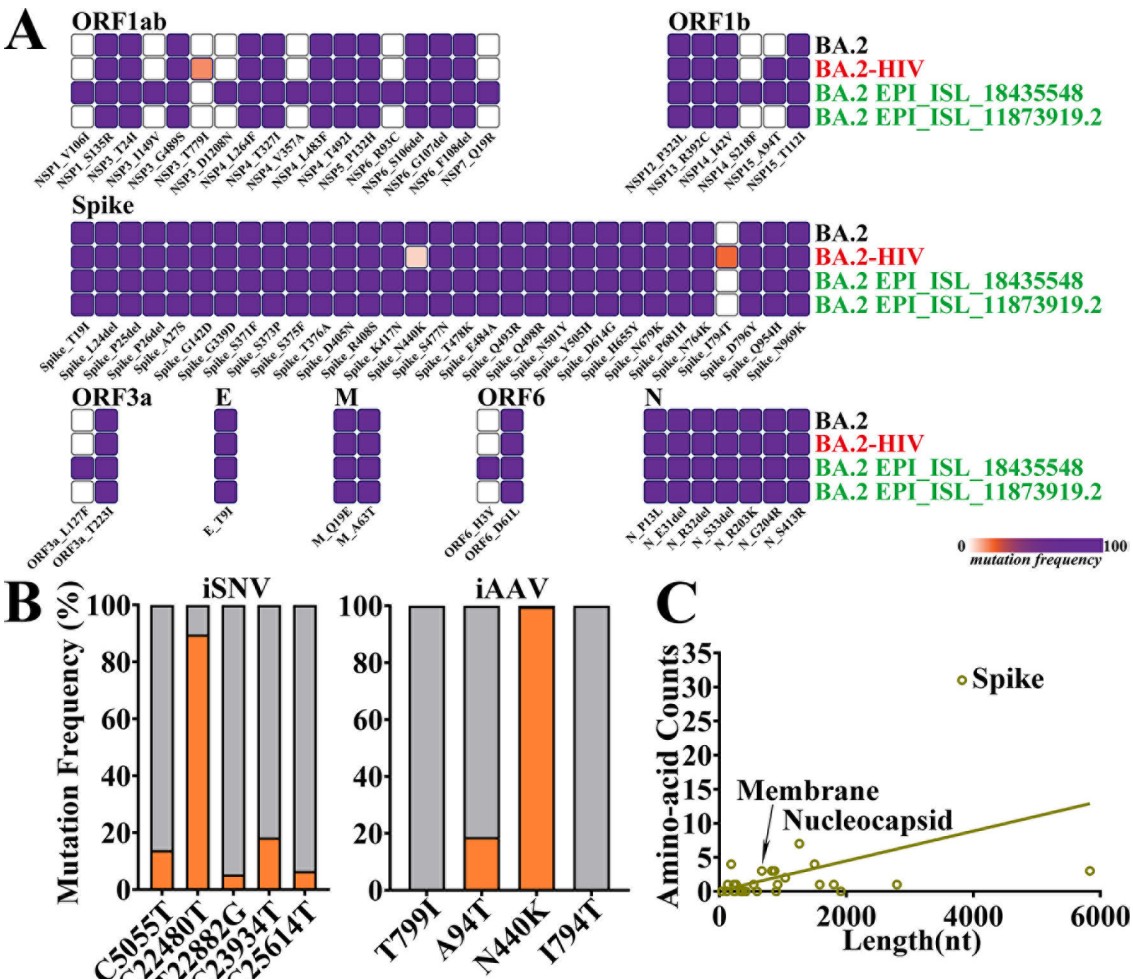

**FIG 4** Viral population dynamics in chronically infected patients. (A) SARS-CoV-2 substitutions identified in chronically infected patients compared with contemporaneous circulating variants: BA.2 reference sequence (black), HIV-SARS-CoV-2 sequence (red), and Jilin Province sample sequence (green). (B) Frequency distribution of intrahost single-nucleotide variants (iSNVs), with annotation of intrahost amino acid variant (iAAV) prevalence in the Chinese population and amino acid (AA) mutation counts across coding regions. (C) Number of amino acid mutations per protein.

virus. Further comparison revealed that the single-round infectivity of the BA.2-Spike-N440 single mutant in Hela-hACE2 cells was significantly lower than that of the double mutant (Fig. 5A), indicating that the mutation at the T794 site may play a dominant role in enhancing infectivity.

To evaluate the impact of the aforementioned mutations on the virus's immune escape ability, we collected serum from healthy individuals who had been previously vaccinated with inactivated vaccines or adenovirus vector vaccines (targeting the wild-type virus). Using a method where pseudotyped vesicular stomatitis viruses were incubated with serially diluted serum before infecting Hela-hACE2 cells, we calculated the 50% neutralizing titer ($NT_{50}$) for each Spike variant. The results showed that relative to the ancestral D614G, the neutralizing titers of all BA.2-related mutant strains were significantly reduced ($P < 0.001$), suggesting that the BA.2 subtype and its mutants all have a certain basis for immune escape. Compared with BA.2-Spike, the neutralizing titer of the BA.2-Spike-N440 single mutant was significantly increased by approximately threefold ($P < 0.001$), and that of the BA.2-Spike-N440/T794 double mutant was significantly increased by approximately twofold ($P < 0.001$). Moreover, there were statistically significant differences in neutralizing titers between the single mutants and the double mutant ($P < 0.001$) (Fig. 5B and C). These results indicate that the mutation at the N440 site can effectively attenuate the virus's immune escape ability, while the

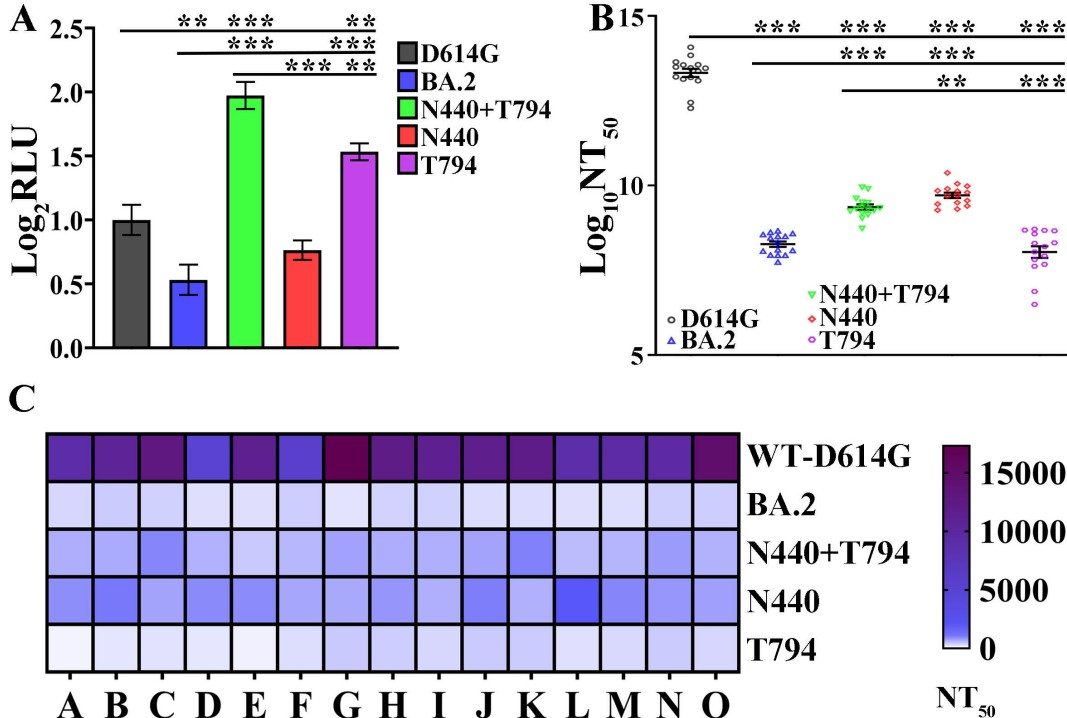

**FIG 5** Neutralization efficacy of inactivated and adenovirus vaccine sera against VSV pseudotyped with Spike protein. (A) Infectivity of spike variants. Relative infectivity was normalized to D614G. Data represent mean ± SD (***$P < 0.001$, $n = 3$). (B) Neutralization titers (50% neutralizing titer [$NT_{50}$]) shown as D614G-normalized relative luminescence units (RLUs) (mean ± SD, **$P < 0.01$, ***$P < 0.001$; $n = 15$ biological replicates per group). (C) Heatmap of individual $NT_{50}$ values across cohorts.

double mutant, due to the presence of the T794 site, shows a rebound in immune escape ability compared to the N440 single mutant.

## DISCUSSION

The findings of this study systematically illuminate the evolutionary trajectory of SARS-CoV-2 in immunocompromised hosts with prolonged infections, shedding critical light on the mechanisms driving viral adaptive mutation and its potential public health implications. By focusing on 24 patients—including an HIV-positive individual with an infection duration exceeding 20 days—we identified key patterns of iSNV accumulation, functional alterations in viral proteins, and evolutionary links to subsequent circulating strains, all of which align with and extend prior research on viral persistence in immuno-deficient populations.

Based on the comprehensive analysis of the above experimental results, it can be seen that the BA.2-Spike-N440/T794 double mutant can significantly enhance the ability to infect host cells. During the infection of SARS-CoV-2 in patients with impaired immune function, host-adaptive mutations in the Spike protein may reduce the virus's immune escape ability through synergistic effects. Among them, the BA.2-Spike-T794 mutation plays a key role in enhancing cell infection efficiency, while the BA.2-Spike-N440 mutation is the core site for reducing the virus's immune escape ability. The balance between the two may affect the replication and transmission potential of the variant strain in the host.

In conclusion, this study reinforces the critical role of immunocompromised hosts in SARS-CoV-2 evolution, demonstrating that prolonged infection drives the accumulation of adaptive mutations with context-dependent effects on infectivity, immune evasion, and cell fusion. The link between the HIV patient's I794T mutation and the later JN.1.16 strain highlights the need for targeted surveillance of immunocompromised

populations, including those with HIV, hematological malignancies, or post-transplant immunosuppression to detect potentially transmissible variants early. Additionally, our findings emphasize that viral evolution in these hosts is shaped by complex trade-offs between fitness traits, underscoring the importance of integrating genomic surveillance with functional assays to assess the public health risk of emerging mutations. Such approaches will be essential for refining public health strategies, particularly as SARS-CoV-2 continues to circulate and adapt in diverse host populations.

## Mutation sites of spike-pseudotyped virus enhance cell-cell membrane fusion capacity

We conducted cell-cell membrane fusion experiments to further explore the functional differences among different mutants and determined the luciferase activity at 0, 6, 12, and 24 h after the superposition of the two types of cells. At 0 h, fusion had not yet started, and there was no difference among the experimental groups. At 6 h, compared with WT-Spike-D614G, all mutants showed significant differences. At 12 h, compared with WT-Spike-D614G, the BA.2 ($P < 0.001$) and BA.2-Spike-N440 ($P < 0.05$) mutants had significant differences, and the difference of the BA.2 mutant was more significant. At 12 h, compared with BA.2, the BA.2-Spike-N440, BA.2-Spike-T794, and BA.2-Spike-N440/T794 mutants all had significant differences, among which the difference in the double-site mutation was more significant. At 24 h, due to the decrease in substrate activity, there was no difference among all experimental groups (Fig. 6A). These results indicate that the BA.2 mutant has the lowest cell-cell membrane fusion capacity, while the cell-cell membrane fusion capacity of the double mutation site is the closest to that of WT-Spike-G614.

To comprehensively characterize the membrane fusion properties of different Spike mutants, in addition to determining luciferase activity, this study also analyzed by observing the size and fluorescence intensity of GFP signal syncytia formed at 24 h. The GFP signal syncytia formed by WT-Spike-G614 were the largest, with the strongest fluorescence signal, while those formed by BA.2 were the smallest, with the weakest fluorescence signal. The GFP signal syncytia formed by BA.2-Spike-N440, BA.2-Spike-T794, and BA.2-Spike-N440/T794 all showed enhanced size and fluorescence intensity compared to BA.2, among which the enhancement effect of BA.2-Spike-N440/T794 was the most significant (Fig. 6B). This result is consistent with the luciferase activity detection results, further confirming the differences in membrane fusion ability among different Spike mutants; that is, the double mutant can significantly restore the membrane fusion function of the virus, making it closer to the level of WT-Spike-G614, while the membrane fusion ability of BA.2 is significantly weaker.

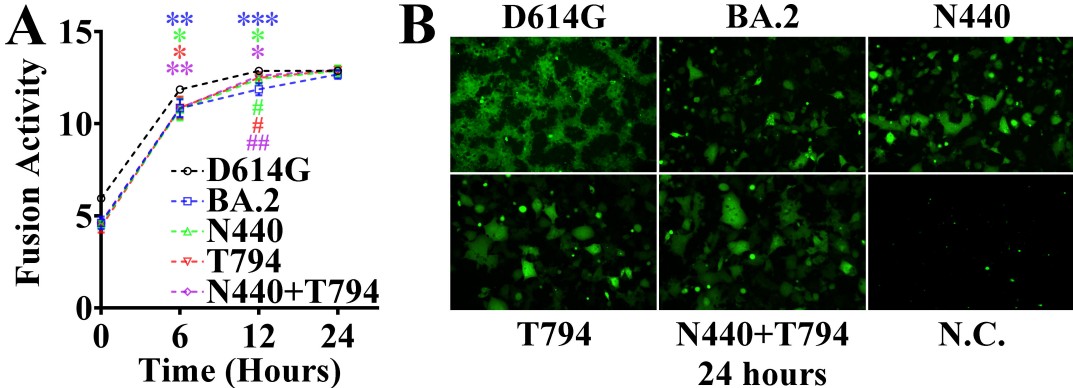

**FIG 6** Spike pseudotype virus mutation site has the ability to fuse cell-cell membranes. (A) Ability of Spike pseudotyped virus mutation sites to fuse and fuse cell-cell membranes at different times. (B) Spike pseudotyped virus mutation site 24 h on cell-cell membrane fusion GFP fluorescence signal.

## Conclusion and perspective

In immunocompromised patients with long-term infection of SARS-CoV-2, the virus often accumulates numerous adaptive mutations due to persistent replication. These mutations may significantly alter its pathogenic characteristics, including infectivity, pathogenicity, and immune escape ability. Focusing on this important scientific issue, this study conducted a retrospective analysis of 24 early patients who tested persistently positive in nasopharyngeal RT-qPCR. The time of symptom onset varied among the included patients, averaging 7–14 days, among which the duration of symptoms in patients with HIV coinfection was significantly prolonged, exceeding 20 days. To explore the drivers of viral adaptive evolution in depth, the research team comprehensively and precisely characterized the mutation status of the viral genome by generating whole-genome ultra-deep sequencing. These infection cases all occurred in April–May 2022, and gene sequencing results showed that most of the infected strains were Omicron variants.

Through comparative analysis with viral sequences from patients with ordinary SARS-CoV-2 infection during the same period, the study found that the viral strains from patients with SARS-CoV-2-HIV coinfection had additional specific mutations, specifically including NSP3-T779I, NSP15-A94T, Spike-N440K, and Spike-I794T. To clarify the biological functions of these mutations, pseudotyped vesicular stomatitis viruses carrying the target Spike mutations were constructed for functional verification. The results showed that the pseudoviruses carrying the Spike-T794 mutation and the Spike double mutation sites had significantly enhanced ability to infect cells, while the Spike-N440 mutation could increase the neutralizing activity of serum against the pseudovirus, implying that this mutation might reduce the virus's immune escape ability. The GFP signal syncytia formed by BA.2-Spike-N440, BA.2-Spike-T794, and BA.2-Spike-N440/T794 all showed enhanced size and fluorescence intensity compared to BA.2, among which the enhancement effect of BA.2-Spike-N440/T794 was the most significant. Further analysis of prevalent strains showed that the proportion of the above mutation sites in the SARS-CoV-2 strains prevalent during the same period was low. However, it is noteworthy that the Spike-T794 mutation was detected in subsequent JN.1.16 clinical samples. This finding suggests that adaptive mutations generated by SARS-CoV-2 during long-term replication may have potential transmission risks.

In sample 2 of the immunodeficient samples shown in Fig. S1, the iAAV not only has a high quantity but also includes mutations with significant functional implications, namely, Spike-N440/T794 double mutations. Subsequent functional experiments confirmed that this double mutation can significantly enhance viral infectivity (Fig. 5A), indicating that these mutations are not random accumulations but are selected by host immune pressure, representing adaptive evolutionary mutations. This further reflects the characteristic of "directional accelerated evolution" of the virus in an immunodeficient environment.

## MATERIALS AND METHODS

### Clinical-sample collection and high-throughput sequencing

Serial samples were collected from the patient periodically from the upper respiratory tract (throat and nasal swab). Nucleic acid extraction was done from 140 M of the sample, using the QIAamp Viral RNA Mini Kit (Qiagen) according to the manufacturer's instructions. All samples were tested for the presence of SARS-CoV-2 with a validated one-step RT-qPCR assay in the Public Health China Clinical Center. Amplification reactions were all performed on a Bio-Rad PCR instrument. Samples with a CT value of ≤36 were considered to be positive. For the library construction, amplified products were fragmented (300–500 bp) and subjected to end repair, dA-tailing, and adaptor ligation using Illumina DNA Prep Kit (Illumina, USA); libraries were purified with AMPure XP Beads (Beckman Coulter, USA) and quantified via KAPA Library Quantification Kit (KAPA Biosystems, USA)

(concentration ≥2 nM). For the high-throughput sequencing, libraries were sequenced on Illumina NovaSeq 6000 platform (Illumina) with PE150 mode (S4 Reagent Kit), cluster density 1,200–1,400 K/mm$^2$, PF cluster % ≥85%. 5. For the read generation, raw signals were converted to base calls via RTA (v.3.4.4); samples were demultiplexed with bcl2fastq v.2.20 (--barcode-mismatches 1); FastQ files were generated and pre-filtered with FastQC (v.0.11.9) (adapter trimming via Cutadapt v.4.0, low-complexity read removal via PRINSEQ-lite v.0.20.4).

## Sequence alignment, genome reconstruction, and variant calling

Sequence reads were first filtered to remove low-quality bases using Trimmomatic (v.0.39) with the following parameters: leading bases with Phred quality of <20 were trimmed (LEADING:20). Trailing bases with Phred qualityof <20 were trimmed (TRAIL-ING:20). A sliding window of four consecutive bases was applied, and sequences were truncated if the average quality within the window was <20 (SLIDINGWINDOW:4:20). Reads shorter than 50 bp after trimming were discarded (MINLEN:50). These trimmed sequences were then aligned to an early Wuhan reference sequence (NC_045512.2) using the BWA aligner tool (v.0.7.17). Secondary quality control of the cleaned reads filtered for low-quality bases was performed using FastQC v.0.11.9. The Clean Reads were aligned to the SARS-CoV-2 reference genome (Wuhan-Hu-1, NC_045512.2) using BWA-MEM (v.0.7.17). The SAM files were optimized using Samtools (v.1.10). Based on the optimized BAM files, virus genome sequence reconstruction was completed in conjunction with Bcftools (v.1.10.2-34) and IGV (v.2.16.0). On the basis of the genome reconstruction, iSNVs were identified using Bcftools mpileup call. Variant calling (hSNP identification): intrahost single-nucleotide variants (hSNPs/iSNVs) were identified via bcftools mpileup -f NC_045512.2. fasta -Q 20 -d 10000 (calculating base pileup with minimum base quality $Q$ ≥20, maximum depth capped at 10,000 to avoid PCR duplicate bias) and bcftools call -m -v (multiallelic calling). To ensure reliability, hSNPs were further filtered with the following criteria: minimum read depth at the variant site ≥30× (only high-confidence sites with sufficient coverage were retained); minimum allele frequency of the variant base ≥5% (to exclude random sequencing errors, typically ≤ 1%); average Phred quality of variant bases ≥20 (error rate ≤1%); and no strand bias (ratio of variant bases in forward vs reverse strands ≥0.3).

## Phylogenetic analysis

Phylogenetic analysis was performed using the maximum likelihood (ML) method. For the sequence data set preparation, high-quality genome sequences of 14 patients (CT <35, sequencing coverage ≥98%) were included, along with representative BA.2 sequences from GISAID. The Wuhan-Hu-1 strain (NC_045512.2) and B.1 strain (PX622476) were used as outgroups. The ML tree was constructed using IQ-TREE (v.2.2.0.3) with parameters.

## Virus isolation and plaque purification

The stock sample solution was inoculated onto a monolayer of Vero E6-TMPRSS2 cells. After 48 h of incubation, the cell culture supernatant was collected and subjected to three to four serial passages. Viral RNA was extracted from each passage using the QIAamp Viral RNA Mini Kit (Qiagen), followed by viral copy number quantification using RT-qPCR to confirm positivity. The virus was purified through a combination of limiting gradient dilution and plaque purification. Briefly, confluent cells were seeded in six-well plates and allowed to form a 90% monolayer. After the culture medium was discarded, the virus was serially diluted 10-fold across five gradients. Then, 300 µL of each diluted virus suspension was added to the wells, while 300 µL of DMEM alone was used as a negative control. Following 1 h adsorption at 37°C, the inoculum was removed. A 2% carboxymethylcellulose solution was gently mixed 1:1 with 2× cell maintenance medium. The mixture was then incubated in a 37°C, 5% $CO_2$ incubator for

48 h. Following incubation, cells were fixed by adding 4–5 mL of 4% paraformaldehyde (PFA) per well, ensuring complete coverage, and incubated at room temperature for 1–2 h. After fixation, plates were gently rinsed under running water to remove residual PFA. Plaques were visualized by staining with 1% crystal violet (300 µL/well) and subsequently photographed. All procedures were conducted under biosafety level 3 containment to ensure safe handling of infectious materials.

## Infectivity of Spike pseudotyped vesicular stomatitis virus with target

Confluent 293T cells were seeded in 12-well plates 24 h prior to transfection. Cells were transfected with 2 µg of pcDNA3. 1-Spike-$\Delta$19 plasmid (or other spike mutant variants) using polyethyleneimine transfection reagent. For pseudovirus infection, VSV-Luciferase-G ($1 \times 10^5$ $TCID_{50}$) was diluted in DMEM supplemented with 10% fetal bovine serum (FBS) to a final volume of 300 µL, thoroughly mixed in a 1.5 mL microcentrifuge tube, and added to the transfected cells. The infection proceeded for 6–8 h at 37°C in a 5% $CO_2$ humidified incubator. Following infection, the culture medium was aspirated and cells were washed twice with phosphate-buffered saline. Fresh DMEM supplemented with 10% FBS (1 mL/well) was added, and cultures were maintained at 37°C in a 5% $CO_2$ humidified incubator for 36 h. The supernatant was then collected and centrifuged at $3,000 \times g$ for 10 minutes at 4°C to remove cellular debris. Purified viral stocks were aliquoted (500 µL/vial) and stored at −80°C for long-term preservation. All plasmids and viral stocks were archived at the Military Medical College repository.

## Neutralizing titer determination

Test sera were heat-inactivated at 56°C for 30 minutes and serially diluted (starting dilution 1:10) in 50 µL volumes. An equal volume (50 µL) of pseudovirus suspension (100 $TCID_{50}$) was added to each serum dilution, thoroughly mixed, and incubated at 37°C for 2 h to allow neutralization. Subsequently, 50 µL of Hela-hACE2 cell suspension ($1-2 \times 10^5$ cells/mL) was added to each well, and plates were incubated at 37°C for 24 h. All samples were tested in duplicate. Each 96-well plate included cell controls (three wells, 50 µL cell suspension + 50 µL DMEM) and pseudovirus controls (three wells, 50 µL cell suspension + 50 µL pseudovirus suspension). After 24 h, relative luminescence units (RLUs) were measured. The neutralization percentage was calculated as [(RLU serum − RLU cell control) / (RLU pseudovirus control − RLU cell control)] × 100%. The serum neutralization titer ($NT_{50}$) was defined as the reciprocal dilution yielding 50% reduction in pseudovirus infection compared to virus controls.

## ACKNOWLEDGMENTS

This work was supported by the National Natural Science Foundation of China (grant no. 8237161174) and the Jilin Provincial Scientific and Technological Development Program (grant no. 212557SF01085113).

Y.L., C.D., and J.Z. performed the sample collection, clinical evaluation, and primary diagnosis of the patient. F.Y. and Q.J. performed the library preparation and sequencing of the samples. Q.J. provided the samples that were purified for toxicity. F.Y. provided a pseudotyped vesicular stomatitis virus experiment with a stinging mutant. X.X., Y.G., X.W., and Z.H. conceptualized the project, provided the overview and guidance, and contributed to writing the manuscript with F.Y. and Q.J.

## AUTHOR AFFILIATIONS

[1]College of Wildlife and Protected Areas, Northeast Forestry University, Harbin, China
[2]Changchun Veterinary Research Institute, Chinese Academy of Agricultural Sciences, State Key Laboratory of Pathogen and Biosecurity, Key Laboratory of Jilin Province for Zoonosis Prevention and Control, Changchun, China
[3]College of Veterinary Medicine, Northeast Agricultural University, Harbin, China

⁴Department of Respiratory Medicine, Second Hospital, Jilin University, Changchun, China

## AUTHOR ORCIDs

Fang Yan  http://orcid.org/0009-0000-3203-1301
Xuefeng Wang  http://orcid.org/0000-0002-3974-3421
Yuwei Gao  http://orcid.org/0000-0002-8278-2418
Jie Zhang  http://orcid.org/0000-0001-6795-8000
Zhijun Hou  http://orcid.org/0000-0002-8704-1651

## DATA AVAILABILITY

Long-read sequencing data that support the findings of this study have been deposited in the National Center for Biotechnology Information Sequence Read Archive database (PX587937, PX587938, PX587939, PX587940, PX587941, PX587942, PX587943, PX587944, PX587945, PX587946, PX587947, PX587948 and PX587949). All data are also available from the corresponding author. Source data are provided with this paper.

## ETHICS APPROVAL

The study was approved by the Ethics Committee of the Academy of Military Medical Sciences (AF/SC-08/02. 448). Written informed consent was obtained from both the patient and his family.

## ADDITIONAL FILES

The following material is available online.

### Supplemental Material

**Figure S1 and Table S1 (Spectrum02432-25-s0001.docx).** Proportion of total iSNVs (%) and intrahost amino acid variants.

### Open Peer Review

**PEER REVIEW HISTORY (review-history.pdf).** An accounting of the reviewer comments and feedback.

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
