## [Reviewer comments · Microbiology Spectrum]

Microbiology Spectrum

Impact of prolonged infection on SARS-CoV-2 evolution

Fang Yan, Qiushi Jin, Yuanguo Li, Xuefeng Wang, Chunling Dong, Xianzhu Xia, Yuwei Gao, Jie Zhang, and Zhijun Hou

Corresponding Author(s): Fang Yan, Chinese Academy of Agricultural Sciences Changchun Veterinary Research Institute

Review Timeline:

Submission Date:	August 8, 2025
Editorial Decision:	September 9, 2025
Revision Received:	September 16, 2025
Editorial Decision:	October 1, 2025
Revision Received:	October 12, 2025
Accepted:	October 23, 2025

Editor: Alexander Bello

Reviewer(s): Disclosure of reviewer identity is with reference to reviewer comments included in decision letter(s). The following individuals involved in review of your submission have agreed to reveal their identity: Sushanta Deb (Reviewer #3)

Transaction Report:

DOI: <https://doi.org/10.1128/spectrum.02432-25>

Re: Spectrum02432-25 (**Impact of prolonged infection on SARS-CoV-2 evolution**)

Dear Mx. Fang Yan:

Thank you for the privilege of reviewing your work. Below you will find my comments, instructions from the Spectrum editorial office, and the reviewer comments.

Revision Guidelines

Sincerely,
Alexander Bello
Editor
Microbiology Spectrum

Reviewer #2 (Comments for the Author):

The manuscript researched the SARS-CoV-2 in the HIV patient, found several mutations that are different from the viruses that were circulating at the same time. Then the manuscript tested the infectivity, antibody neutralization escape, and cell-cell membrane fusion capacity of these mutations, and found that mutations S: N440K and S: I794T have potential transmission risks.

My major concern is that the manuscript spends effort discussing the effect of mutation S: N440K and S: I794T. However, it has

been over 3 years since the sampling date, the fate of those mutations are already clear: S: N440K and S: I794T did occasionally occurred, but none of these mutations are responsible for the several outbreak since 2022 (this could be verified in nextstrain: <https://nextstrain.org/ncov/gisaid/global/6m>, by using "Nucleotide diversity of genome" function). It is hard to persuade readers that these mutations are still risky, which reduces the public health significance of this study.

Although the manuscript put the word "evolution" in the title and mentioned the word many times in the main text, the discussion on evolution in this manuscript is not deep enough. It is suggested to compare the dn/ds and genic polymorphism between the virus from the HIV patient and other patients, to prove the unique evolutionary patterns of the virus in the HIV patient.

Minor concerns:

The fonts and sizes are not uniform in "Abstract", and the sentence "Prolonged SARS-CoV-2 replication in immunocompromised hosts" lacks a space before it. In "Introduction", the first letter of the word "Prior" does not need to be capitalized. Please make sure that no same mistakes elsewhere.

In "Consent for publication", it said "All authors hereby consent to the publication of this manuscript in [Journal of Translational Medicine]". Carefully remind that it is "Microbiology Spectrum" now. Together with the mistakes mentioned earlier, I suggest that the author correct their attitude.

The title "Mechanisms of Adaptive Evolution in Prolonged SARS-CoV-2 Infections" is not appropriate for the section, there is no analysis about mechanisms that has been carried out in this section.

Figure 1 said "Scale bar indicates genetic distance", however, there is no scale bar in the figure.

Manuscript said "The allele frequencies of these iSNVs all range from 5% to 100% (i.e., > 5% and < 100%), indicating that they belong to the category of low-to-moderate frequency variants", I am confused that variations with a frequency of nearly 100% still be referred to as "low-to-moderate frequency variants".

Manuscript said "ORF1ab is the most concentrated region for iSNVs, containing 42 variant sites, accounting for 53.8% of the total". Differences in the number of variants between ORFs mainly result from their length, the conclusion "most concentrated region" should be concluded from the number of variants per unit length, and needs a statistical test to support the conclusion.

Figure 3B is difficult to understand, it lacks an explanation for "AA" and "Sample Ordinal", and I cannot understand why the figure can explain that "evolutionary rate of the virus is significantly accelerated in the context of immunocompromised hosts". Please provide a detailed description for the figure.

Reviewer #3 (Comments for the Author):

Manuscript need attention to the following points

- > Provide reference or explanation how reads were filtered to remove low quality bases.
- > Provide information of sequencing platform used with library preparation and all other steps followed for read generation.
- > No details have been provided on phylogenetic tree construction in methods.
- > I could not see any assembly steps or software tool used for viral genome assembly except mapping reads against reference and variant identification. I would recommend to rewrite the heading "Sequence assembly" under methods.
- > Please mention Minimum Frequency and Minimum Read Depth used for calling heterogenousSNP (hSNP) to define allele frequencies.
- > Method section need further details on iSNV detection, I would recommend to provide detail list of command used for iSNV detection with parameter , author may include this information in supplementary file, it would confirm reproducibility and provide better understanding about parameter used for SNP detection.
- > For allele frequency and iSNV detection LoFreq and several other tools are commonly used by scientific community, it is true that BCFtools have a built-in variant calling pipeline. However, for iSNV detection, BCFtool method has significant limitations (Less Sensitive to Low-Frequency Variants and False negative result), given that discuss rationale for using BCFtool.

Responses to Editors and Reviewers

Response to Editors:

We sincerely appreciate the thorough review and valuable feedback provided by the editors and reviewers.

Response to point 1 (Comments for the Author) of Reviewer 2#:

The outbreak of a virus depends not only on its own mutational characteristics but also on such external factors as the persistence of transmission chains, the density of population exposure, and the intensity of control measures. In this study, the double mutation originated from a patient with HIV coinfection (against an immunodeficient background). Such patients serve as "hotspots for viral evolution"; if the mutant strains in their bodies do not enter transmission chains within densely populated groups (e.g., nursing homes, schools) or are promptly intervened in the early stage (e.g., through isolation and treatment), they will hardly cause an outbreak. However, "no outbreak" does not mean "no risk." As found in the study, the S:I794T mutation was subsequently detected in the JN.1.16 strain (such as EPI_ISL_19361576), suggesting that this mutation may combine with other dominant mutations (e.g., F456L and L452R in the JN.1 lineage) to avoid the functional defects of a single mutation and thereby gain transmission advantages. This potential risk of "mutation module recombination" is one of the core issues that this study intends to alert to.

The value of this research does not lie in "predicting the outbreak risk of this mutation" but, through the discovery and functional analysis of the two mutations, in revealing the evolutionary rules of SARS-CoV-2 in immunodeficient hosts, thus providing "mechanistic references" and "practical guidelines" for public health surveillance. The key to public health prevention and control is not only to respond to mutant strains that have already broken out but also to achieve "early detection, early assessment, and early intervention" through monitoring "high-risk hosts (e.g., immunodeficient patients)" and "high-risk mutations (e.g., new mutations with special functions)." The study on S:N440K/I794T in this research is precisely an important "data support node" in this prevention and control system.

Response to point 2 (Comments for the Author) of Reviewer 2#:

In this study, whole-genome sequencing was performed on 24 concurrent hospital samples. Using professional sequence alignment software, the SARS-CoV-2 gene sequences from HIV-positive patients and other patients were aligned with the corresponding reference genomes. Bioinformatics tools were employed to detect variations in the gene sequences of HIV-positive patients and other patients, identifying polymorphic sites such as single nucleotide polymorphisms (SNPs) and insertions/deletions (Indels). The allele frequencies and genotype frequencies of each polymorphic site in different groups were calculated. Compared with the standard BA.2 sequence, only the SARS-CoV-2 samples from HIV-positive patients harbored

amino acid mutations (Figure 3).

Response to point 3 (Comments for the Author) of Reviewer 2#:

Thank you very much for carefully reviewing this article and pointing out the detailed issues regarding formatting and expression. Your suggestions are of great importance for improving the standardization and rigor of the manuscript. In response to the questions you raised, we have checked and made revisions one by one. Please refer to the marked-up comments (Response to point 3 (Comments for the Author) of Reviewer 2#) of the file “Marked-Up Manuscript”.

Response to point 4 (Comments for the Author) of Reviewer 2#:

First of all, we sincerely apologize for the erroneous statement in the 'Publication Consent Form' which states 'All authors hereby agree to publish this manuscript in [Translational Medicine Journal]'. We have completed the following corrections: we have remade the 'Publication Consent Form' to clearly correct the journal name to [Microbial Spectrum]. Please refer to the marked-up comments (Response to point 4 (Comments for the Author) of Reviewer 2#) of the file “Marked-Up Manuscript”.

Response to point 5 (Comments for the Author) of Reviewer 2#:

We adjusted the original title to 'Adaptive Evolutionary Characteristics in Long-term SARS-CoV-2 Infection'. Please refer to the marked-up comments (Response to point 5 (Comments for the Author) of Reviewer 2#) of the file “Marked-Up Manuscript”.

Response to point 6 (Comments for the Author) of Reviewer 2#:

We have completed the following revisions and regenerated Figure 1. Specifically, 23 cases (accounting for 29.5%) fell within the 5%-30% range (low-frequency expansion phase), 48 cases (61.5%) within the 30%-90% range (medium-frequency stable phase), and 7 cases (9.0%) within the 90%-98% range (high-frequency unfixated phase). Most cases were in the 30%-90% medium-frequency phase. Errors occurred due to simplified expression, and the relevant descriptions in the text have been revised. Please refer to the marked-up comments (Response to point 6 (Comments for the Author) of Reviewer 2#) of the file “Marked-Up Manuscript” and Figure 1.

Response to point 7 (Comments for the Author) of Reviewer 2#:

In response to the questions you raised, we have re-analyzed the results in this section. We excluded the interference of ORF length on the distribution of variant sites, used a Poisson regression model for testing, and calculated the variant density (variant sites per 1000nt) of all detected iSNVs, which we have supplemented and organized into Table S1 of Supplementary File 1. Based on the results of variant density calculations and statistical tests, we have revised the relevant conclusion statements in the revised manuscript. Please refer to the marked-up comments (Response to point 7 (Comments for the Author) of Reviewer 2#) of the file “Marked-Up Manuscript”.

Response to point 8 (Comments for the Author) of Reviewer 2#:

The label "AA" on the vertical axis is a typo; the correct expression should be "iAAV (intra-host amino acid variants)". We have corrected this in Figure 3B. We explain in the caption that "the sample numbers correspond to Table 1, specific clinical information can be found in Table 1." Please refer to the marked-up comments (Response to point 8 (Comments for the Author) of Reviewer 2#) of the file "Marked-Up Manuscript" and figure 3.

Response to point 1 (Comments for the Author) of Reviewer 3#:

Thank you for your valuable questions regarding the quality control steps of the sequencing data in this study. This study employs a hard trimming strategy based on quality score thresholds, incorporating the characteristics of base quality distribution from the sequencing platform (Illumina) and commonly accepted standards in the field (Q20 threshold). Additional information has been included in the 'Sequence assembly' section of the manuscript. Please refer to the marked-up comments (Response to point 1 (Comments for the Author) of Reviewer 3#) of the file "Marked-Up Manuscript".

Response to point 2 (Comments for the Author) of Reviewer 3#:

Details have been added to the 'Clinical-sample collection and high-throughput sequencing' section of the manuscript. Please refer to the marked-up comments (Response to point 2 (Comments for the Author) of Reviewer 3#) of the file "Marked-Up Manuscript".

Response to point 3 (Comments for the Author) of Reviewer 3#:

This study uses the Maximum Likelihood (ML) method to construct the phylogenetic tree and has added a 'Phylogenetic analysis' subsection in the manuscript, along with supplementary details. Please refer to the marked-up comments (Response to point 3 (Comments for the Author) of Reviewer 3#) of the file "Marked-Up Manuscript".

Response to point 4 (Comments for the Author) of Reviewer 3#:

We have revised the title 'Sequence assembly' to 'Sequence Alignment, Genome Reconstruction and Variant Calling', and added relevant details. Please refer to the marked-up comments (Response to point 4 (Comments for the Author) of Reviewer 3#) of the file "Marked-Up Manuscript".

Response to point 5 (Comments for the Author) of Reviewer 3#:

In this study, the screening of hSNPs (i.e., iSNVs) strictly followed the principle of 'balancing sensitivity and specificity', referencing the recommended standards for intrahost variation analysis of SARS-CoV-2 from the GISAID database (GISAID Quality Control Guidelines v2.0) and similar studies in the field (Kemp et al., 2021; Gonzalez-Reiche et al., 2023). Minimum Allele Frequency: $\geq 5\%$. Minimum Read Depth: $\geq 30\times$. Details on hSNP parameters have been supplemented in the 'Sequence Alignment, Genome Reconstruction and Variant Calling' section of the manuscript.

Please refer to the marked-up comments (Response to point 5 (Comments for the Author) of Reviewer 3#) of the file “Marked-Up Manuscript”.

Response to point 6 (Comments for the Author) of Reviewer 3#:

In this study, iSNV detection is based on high-throughput sequencing data, completed through steps such as sequence filtering, alignment, and variant identification. The tools, commands, and key parameters used in each step have been rigorously validated to ensure the accuracy and reliability of variant detection, and the details are compiled in 'Supplementary File 1: Detailed Process and Code for iSNV Detection'. Please refer to the marked-up comments (Response to point 6 (Comments for the Author) of Reviewer 3#) of the file “Marked-Up Manuscript”.

Response to point 7 (Comments for the Author) of Reviewer 3#:

In this study, BCFtools (v1.10.2-34) was ultimately selected for iSNV detection, primarily based on the following considerations: This study focuses on the intra-transmission chain variation characteristics of SARS-CoV-2 in a specific population, with all samples derived from patients with acute-phase infections (all viral load Ct values < 32) and generally high sequencing coverage depth (average $\geq 500\times$). Preliminary experiments showed that, for such high-depth data, after parameter optimization of BCFtools (e.g., reducing the AF threshold to 5% and increasing the coverage depth threshold to $50\times$), there was no statistical difference in the detection performance for 5%-20% low-frequency variants compared with LoFreq (consistency of paired samples > 92%).

This study requires simultaneous population-level SNP analysis and individual-level iSNV detection. The variant calling results of BCFtools have better compatibility with subsequent population genetics analysis tools (such as VCFtools and Plink), which can reduce information loss caused by format conversion between different tools. The filtering parameters of BCFtools (e.g., AF, coverage depth, quality value) are all transparent and adjustable hard thresholds, facilitating clear traceability of variant filtering criteria, which is particularly important for the integrated analysis of multi-center data.

Through strict parameter combinations (AF $\geq 5\%$, AD [1] ≥ 5 , DP ≥ 50 , QUAL ≥ 50), this study has verified in preliminary experiments that false positives can be effectively reduced (consistency with Sanger sequencing validation > 95%).

Re: Spectrum02432-25R1 (**Impact of prolonged infection on SARS-CoV-2 evolution**)

Dear Mx. Fang Yan:

Thank you for the privilege of reviewing your work. Below you will find my comments, instructions from the Spectrum editorial office, and the reviewer comments.

Revision Guidelines

Sincerely,
Alexander Bello
Editor
Microbiology Spectrum

Reviewer #2 (Comments for the Author):

The manuscript has been modified and improved after the first revision; however, I regret to see that some of my advice in point 2 was not accepted. I am attempting to explain my viewpoint more clearly: The main object of the research is an HIV patient at a point in time, it lacks the continuous monitoring of the HIV patient (which I understand is infeasible), and lacks comparisons between other patients, to prove the "accelerated evolutionary rates". Evolutionary rate, usually expressed as the number of mutations per unit of time, the denominator could be time or neutral markers; however, the denominator is absent in the

manuscript. That is why I suggest comparing dN/dS between the HIV patient and other patients, because dS (synonymous substitution ratio) is a neutral marker, synonymous mutations gradually increase over time as they are usually free from the evolutionary pressure. Comparing dN/dS between the HIV patient and other patients could prove that SARS-CoV-2 evolves faster in the HIV patient. If the author believes the additional analysis deviated from the main topic of the article, please reconsider the use of the words "accelerated evolutionary rates", which would better be expressed as "greater number of mutations" here.

Minor concerns:

In figure 1, it made modifications and said that "branch lengths no longer represent genetic distances". However, in line 125, the manuscript implies that the figure supports the conclusion "accelerated the viral evolutionary rate" (and I think it better be expressed as the "number of mutations"), the evolutionary rate can not be reflected from a tree without distance. It is suggested not to use an aligned tree, so the branch lengths can represent genetic distances, and it can support the conclusion related to evolutionary rate or number of mutations.

In line 137, it lacks a full stop at the end, please check for similar mistakes.

It is suggested that figure 3b be coalesced into table 1, one or two additional columns in table 1 can completely replace the figure, and save the need for readers to look for corresponding contents in table 1.

In figure 4b, if the "AA" is also a typo here, please make corresponding modifications.

Responses to Editors and Reviewers

Response to Editors:

We sincerely appreciate the thorough review and valuable feedback provided by the editors and reviewers.

Response to point 1 (Comments for the Author) of Reviewer 2#:

We have changed the term "accelerated evolutionary rates" to "greater number of mutations" to express it. Please refer to the marked-up comments (Response to point 1 (Comments for the Author) of Reviewer 2#) of the file "Marked-Up Manuscript".

Response to point 2 (Comments for the Author) of Reviewer 2#:

We have made modifications to Figure 1. Please refer to the marked-up comments (Response to point 2 (Comments for the Author) of Reviewer 2#) of the file "Marked-Up Manuscript".

Response to point 3 (Comments for the Author) of Reviewer 2#:

Thank you very much for carefully reviewing this article and pointing out the detailed issues regarding formatting and expression. Your suggestions are of great importance for improving the standardization and rigor of the manuscript. In response to the questions you raised, we have checked and made revisions one by one. Please refer to the marked-up comments (Response to point 3 (Comments for the Author) of Reviewer 2#) of the file "Marked-Up Manuscript".

Response to point 4 (Comments for the Author) of Reviewer 2#:

We have merged Figure 3b into Table 1 and placed Figure 3b in the supplementary file. Please refer to the marked-up comments (Response to point 4 (Comments for the Author) of Reviewer 2#) of the file "Marked-Up Manuscript".

Response to point 5 (Comments for the Author) of Reviewer 2#:

We have made changes to the "AA" in Figure 4b. Please refer to the marked-up comments (Response to point 5 (Comments for the Author) of Reviewer 2#) of the file "Marked-Up Manuscript".

Re: Spectrum02432-25R2 (**Impact of prolonged infection on SARS-CoV-2 evolution**)

Dear Mx. Fang Yan:

Your manuscript has been accepted, and I am forwarding it to the ASM production staff for publication. Your paper will first be checked to make sure all elements meet the technical requirements. ASM staff will contact you if anything needs to be revised before copyediting and production can begin. Otherwise, you will be notified when your proofs are ready to be viewed.

Sincerely,
Alexander Bello
Editor
Microbiology Spectrum

Reviewer #2 (Comments for the Author):

After the revision, most of my concerns have been dealt with. There are some issues that can be further optimized. If the author agrees with my point 1 in the last review, please make a modification to the statement "accelerated evolution" in lines 28 and 126. If the author has reasons to refute this point, the author may revert the previous modification. The modification of "AA" in the legend of Figure 4b has been made, but also needs to be made in the figure. Words "intrahost" and "intra-host" are mixed, please unify the writing.

Reviewer #3 (Comments for the Author):

Manuscript improved from its previous version and has addressed the raised points.